# Advancements in Wearable Antenna Design: A Comprehensive Review of Materials, Fabrication Techniques, and Future Trends in Wireless Communication

**DOI:** 10.3390/mi16091028

**Published:** 2025-09-08

**Authors:** Zhikai Cao, Mai Lu

**Affiliations:** Key Laboratory of Opto-Electronic Technology and Intelligent Control of Ministry of Education, Lanzhou Jiaotong University, Lanzhou 730070, China; czk19988@163.com

**Keywords:** wearable antenna, material selection, miniaturization technology, application

## Abstract

With the continuous development of wireless communication technology, the demand for wearable communication devices has rapidly increased. The antenna is one of the key components in communication devices, directly affecting the performance of wearable communication devices. As a result, wearable antenna design has become a research hotspot in recent years. Wearable antennas are widely used in various fields of daily life, including healthcare, sports and entertainment, the internet of things (IoT), and military positioning. In the last decade, related researchers have studied wearable antennas from various perspectives, and this paper summarizes the design and fabrication of wearable antennas more comprehensively and systematically. This review covers material selection, manufacturing techniques, miniaturization technologies, and performance metrics, while addressing key design considerations. It also highlights recent research, applications in critical fields, and future development trends, offering valuable insights for the design and study of wearable antennas.

## 1. Introduction

Since the fourth- and fifth-generation wireless communication technologies have entered the field of wearable devices, the applications of wearable devices have been greatly expanded. Fourth generation (4G) technology, with its high-speed data transmission capability, supports the applications of wearable devices in health detection [1], motion tracking [2], information delivery, and location-based services [3], which are accelerated by the development of 5G technology [4].

According to Grand View Research’s “Wearable Technology Market Share & Trends Report, 2030”, the global wearable devices market reached approximately $45 billion in 2023, with smartwatches and fitness trackers dominating market share. The industry is projected to grow at a CAGR of 18.1%, reaching $150 billion by 2030. Rising income levels and declining prices are driving market expansion in the Asia–Pacific region, particularly China, India, and Japan.

Similarly, the “China Smart Wearables Industry Market Competitive Landscape and Investment Outlook Report 2024–2030” by Huajing Industry Research Institute indicates that China’s consumer-grade smart wearables market grew from RMB 21.26 billion to RMB 93.47 billion over the past five years, with a CAGR exceeding 20%. The report forecasts continued growth, with a CAGR above 10% over the next five years. Wearable devices include devices that are worn directly on the body as well as devices that can be integrated into clothing and accessories [5]. These devices communicate with other devices through wireless communication modules. Among these modules, the antenna plays a crucial role. It improves the efficiency of the entire communication system. In the development of wireless communication and biocompatibility technology, the human body as a kind of electromagnetic wave “conductor” will have an impact on the radiation characteristics of the antenna, which will be reflected in the antenna’s efficiency, frequency response, gain, radiation direction map, and so on. The dielectric constant and loss factor of human body tissues vary across different frequency bands. Generally, a higher dielectric constant in human body tissues leads to a reduction in the antenna’s radiated power. It can also change the direction of the antenna’s radiation. On the other hand, factors such as the antenna’s distance from the human body, its orientation, and the way it contacts the body will affect the antenna’s radiating performance. The safety of wearable antennas mainly involves the influence of electromagnetic radiation, thermal effects, and potential biological effects, Wearable antennas primarily generate radiofrequency electromagnetic radiation, which was designated as a potential category 2B carcinogen by the International Agency for Research on Cancer as early as 2011. Therefore, the design should follow the international safety standards for electromagnetic radiation, should optimize the antenna’s power and radiation pattern, and should ensure that the device’s SAR value, RF exposure, and temperature rise are within the safety range. At the same time, rigorous safety assessments and validation should be conducted to ensure that long-term use will not adversely affect human health. In the future, with the development of new technologies (e.g., 5G, 6G) and new materials, further research and testing are still needed to assess the safety of wearable devices.

Wearable antennas are widely used in frequency bands, such as industrial, scientific, medical (ISM) (2.4 GHz–2.4835 GHz, 5.725 GHz–5.875 GHz) [6], and ultra-wideband (UWB) (3.1 GHz–10.6 GHz) [7]. For wearable device applications, the design and fabrication of antennas need to consider factors such as human wearing comfort, safety, fabrication cost, and environment (temperature, humidity, etc.), among other factors. Moreover, the physical space constraints of wearable devices, the need for wide bandwidth, reconfigurability, high directionality, stable efficiency, and multi-band operation are more challenging than traditional antenna design [8].

## 2. The Evolution of Wearable Antenna Technology

The earliest wearable devices can be traced back to the 1960s and 1970s. In the 1960s, the U.S. military developed an early gait monitor to track the activities of soldiers. This device, despite its simplicity, marked the beginning of wearable technology, especially in the areas of motion and biomonitoring. In the 1980s, with the development of computers, computer scientists and electronic engineers began to develop more advanced wearable computers. By the 1990s, wearable technology had initially developed and started transitioning to practical applications. Steve Mann introduced a head-mounted wearable computing device that could superimpose computer-generated images onto the user’s field of view, making it one of the earliest applications of augmented reality technology. With the continuous development of wearable technology, more and more wearable devices have been developed, such as Polar’s heart rate monitor, which enables athletes to monitor their heart rate in real time, and Seiko’s smartwatch, which integrates data storage and management functions to store information, such as addresses, schedules, and notes. This watch represents one of the earliest forms of smartwatches; in 2013, Google launched Google Glass, the first smart glasses to become widely known to the public. Along with the development of wearable devices, the study of wearable antennas integrated in the devices has become more and more intensive.

In 1999, P. Salonen et al. proposed a planar inverted-F antenna for wearable devices in their paper, which generates resonance in a dual band by etching a U-shaped slot [9]. Subsequently, Salonen and Rantanen proposed a dual-band flexible antenna for smart clothing [10] and Salonen et al. proposed a Bluetooth antenna [11] in 2001, respectively; Salonen’s research has pushed the antenna design towards miniaturization and wearability. In 2003, a flexible wearable antenna using a flexible conductive fabric and felt composition was proposed by M. Tanaka and Jae-Hyeuk Jang, which can be integrated into clothing or hats [12]. P. Salonen’s research laid the foundation for the study of wearable antennas, and also inspired researchers to demand broadband and multi-band for wearable antennas. Subsequent researchers further explored wearable antennas with flexible, miniaturized, circularly-polarized, and ultra-broadband characteristics to adapt to the complex electromagnetic environment of the human body and special application scenarios. Early research began to focus on how to design antennas that support multiple frequencies, especially for application scenarios such as Bluetooth, Wi-Fi, and GPS. Researchers tried to innovate in the physical design of antennas, such as by adjusting the antenna size and shape to achieve multi-frequency response. This period of time also coincided with the time when research on flexible antennas was just beginning, and researchers began to experiment with materials such as thin-film technology and conductive polymers to enable antennas to adapt to the curved surfaces and dynamic deformation requirements of wearable devices, taking into account the flexibility required for wearable antennas [13].

The Federal Communications Commission (FCC) formally authorized the implementation of ultra-wideband (UWB) technology in civilian communications through the publication of ET Docket 98–153 in 2002. This authorization stipulated that the primary civilian frequency band for UWB is 3.1–10.6 GHz, with a bandwidth of ≥500 MHz or ≥20% relative bandwidth, and a transmission power not exceeding −41.3 dBm/MHz EIRP. This definition and standard have remained pivotal references for global UWB technology to the present day. Between 2004 and 2006, researchers gradually explored the application of UWB technology in wearable antennas. In the following years, research on UWB antenna design, human signal propagation characteristics, health monitoring, and localization systems was further developed. A UWB antenna brings high bandwidth, low power consumption, and short pulse transmission capability, which is widely used in radar detection, health sensors, and precise localization systems. Considering the miniaturization of wearable antennas, a low-profile wearable electromagnetic bandgap antenna was designed in 2004, which was the first time that an electromagnetic bandgap structure was applied to wearable antennas [14]. In 2007, further research on a dual-band textile antenna containing EBG surfaces was carried out by Shaozhen Zhu and Richard Langley, which adopted a flexible material commonly used in duty manufacturing and integrated the EBG structure, which greatly reduced the back radiation generated by the antenna and could operate in two frequency bands, demonstrating that a dual-band textile antenna integrated with a bandgap surface can be placed on the body and provide as good a performance as conventionally fabricated antennas [15].

The rapid development of flexible materials and electronics in 2010 has prompted researchers to utilize flexible materials, such as metallic nanowires, conductive fibers, and carbon-based materials (e.g., graphene), to engineer antennas that are suitable for wearable applications. These antennas have the capacity to be embedded in clothing and accessories, as well as to be applied to the skin with a certain degree of flexibility and comfort. In addition to their fundamental communication functionality, a growing number of wearable devices are being equipped with an array of features, including heart rate monitoring, GPS positioning, and temperature sensing. This trend underscores the necessity for antennas to be designed with greater diversity to accommodate these multifaceted applications. The advent of material science has paved the way for the development of ultra-thin and transparent antennas. This development is particularly evident in wearable devices, such as smart clothing, smart glasses, and smart bracelets, where transparent antennas can be seamlessly integrated into the device design with minimal or no impact on the device’s appearance [16]. The integration of transparent antennas into wearable devices not only enhances their aesthetic appeal but ensures efficient wireless performance. Antenna design has gradually evolved to accommodate a variety of wireless communication technologies, such as 5G, Bluetooth 5.0, and Wi-Fi 6, in order to meet the growing demand for data transmission speeds. Advances in printing technology have further enabled the integration of antenna designs into textiles and skin patches, facilitating the development of more sophisticated and multifaceted wearable devices. The integration of textiles, fabrics, and skin sensors, among others, with antennas has contributed to the development of more comfortable, discreet, and fully functional wearable devices.

Since 2020, with the gradual commercialization of 5G technology, antenna design in the 5G communication band has increasingly become a central focus in the research and development of wearable devices. Additionally, it is imperative to consider the influence of the human body on wireless signals. The enhanced specifications for wearable antennas, encompassing parameters such as frequency response, gain, and radiation efficiency, have prompted the ongoing optimization in their design.

## 3. The Design and Processing Technology of Wearable Antennas

### 3.1. Materials Utilized in the Fabrication of Wearable Antennas

In contrast to conventional antenna design, which prioritizes performance and mechanical strength, wearable antenna design places greater emphasis on flexibility and wearability. Additionally, textile materials are being employed with increasing frequency in antenna design and fabrication due to their exceptional bendability, reduced weight, and lower dielectric constant. The performance of wearable antennas is influenced by two primary factors: the conductive materials (radiating elements) and the non-conductive materials (substrate materials). The selection of conductive materials depends on their electrical conductivity, while non-conductive materials are chosen based on dielectric properties, flexibility, comfort, and compatibility with the human body. Therefore, wearable antenna designs aim to balance antenna performance with human wearability. The goal is to create a high-gain, high-efficiency, safe, and comfortable antenna.

#### 3.1.1. Conductive Materials

The materials used in the radiating elements of antennas can be broadly classified into three groups. The first category is metal-based materials such as copper [17] and aluminum [18,19], which have good conductive properties and moderate cost. The second category includes conductive fibers [20] and metal wires [21], integrated into the substrate via weaving, sewing, or felting [22]. The third involves conductive ink with metal nanoparticles [23,24], printed on a dielectric substrate using inkjet or screen printing [25] to form conductive paths. Metal-based materials exhibit high conductivity and stable electrical properties, but they are relatively heavy and lack flexibility. Conductive fibers and metal wires offer good flexibility, can be integrated into wearable devices, and are lightweight; however, they possess lower conductivity and limited long-term durability. Conductive inks provide manufacturing flexibility, support miniaturization, and enable the formation of complex patterns, yet their conductivity and mechanical durability are relatively limited.

As shown in Figure 1, a miniaturized flexible antenna based on highly conductive graphene assembly film (GAF) was proposed by Zhang et al. [26] as a wireless wearable sensor, which has a size of 50 mm × 50 mm, operates at 3.13–4.42 GHz, and achieves tensile and compressive bending strain sensitivities of 34.9 and 35.6, respectively, and still has good stability after 100 bending tests. Xiaoyou Lin’s research team used an ultrathin thermally bonded film to fix a radiating element made of copper–polyester taffeta fabric on the fabric substrate by ironing the film. The copper–polyester taffeta radiating element was fixed to the fabric substrate with an ultrathin thermally bonded film, and the film was melted by ironing to bond the two different materials. The compact antenna designed by Lin et al. [27] achieved an ultra-wide operating bandwidth of 109% in the 1.198 GHz to 4.055 GHz range. As shown in Figure 2, Giorgio Montisci et al. proposed a bendable microstrip patch antenna based on transparent conductive films (TCF), which addresses the issue of low radiation efficiency in traditional transparent antennas in the S-band (2.15 GHz) by supporting a 3D-printed PETG frame on a 0.2 mm thick polyethylene terephthalate (PET) substrate. The simulation results indicated that, when the surface resistance of the TCF is between 2 and 10 Ω/sq, the radiation efficiency can reach 72.3%, with a gain of 5.3 dBi. Compared to transparent flat antennas, the simulated efficiency of this antenna at 2.15 GHz is at least four times higher [28].

Table 1 lists the parameters of the conductive materials used in each reference.

#### 3.1.2. Substrate Materials

Conventional antennas are fabricated using printed circuit board (PCB) technology with rigid FR4 substrates [29], other glass-fiber polymer materials [30], and ceramic materials (e.g., Rogers, Taconic) [31,32]. These materials offer the advantage of good rigidity but also present disadvantages in terms of poor elasticity and high cost.

The resulting antennas are challenging to integrate into clothing. When stretched or bent by external forces, the antennas may deform or even break, thereby impairing their performance. The advent of flexible materials has provided a solution to this issue. Flexible material substrates demonstrate favorable flexibility, sufficient gain, and stable radiation patterns in bending tests, and flexible substrates are generally lighter than rigid substrates, which helps reduce the overall weight of the device, especially for devices that need to be worn for long periods of time. However, the low dielectric constant and dielectric loss of flexible materials necessitate the use of larger dimensions to ensure the efficiency of the antenna under human wear. Examples of such materials include denim [33,34], felt [35], silk [36], rubber [37], and polyimide (PI) [38,39]. Polyethylene terephthalate (PET) [40,41], polydimethylsiloxane (PDMS) [42,43], and polytetrafluoroethylene (PTFE) [44,45] are commonly employed as backing materials in flexible antennas, offering excellent thermal and electrical stability. Additionally, some researchers have utilized specialized composite materials and have designed antennas with noteworthy performance [25,46]. The customization of composites for electromagnetic properties can be achieved by selecting the appropriate composition. This enables the design of antennas to meet specific frequency and bandwidth requirements. Through the optimization of composite material composition, the radiation efficiency of the antenna can be enhanced, thereby mitigating signal loss.

As illustrated in Figure 3a, one study proposed chitosan, an environmentally friendly material, as a substrate material for a plastic-free antenna. This antenna was designed using the inkjet printing technique and exhibited a compact size, good flexibility, and a low specific absorption rate. The radiating part of the antenna was made of biocompatible silver nanoparticle ink [47]. As Figure 3b indicates, Hattan F. Abutarboush proposed a unique synthetic paper substrate (Teslin) flexible antenna with exceptional flexibility and a reduced fabrication cost [48]. As illustrated in Figure 3c, Younes Siraj et al. presented a terahertz slit patch antenna based on a polyimide substrate with an integrated 6 × 5 array of metamaterials operating at 3.62 THz. The gain is enhanced from 3.89 dB to 6.04 dB by optimizing the metamaterials, which significantly improves the impedance matching and radiation performance [49]. As illustrated in Figure 3d, Zhihao Zhuang et al. proposed a wearable antenna sensor based on ePDA/SiO_2_ nanowalls, using PDMS as a flexible substrate, operating at a 2.45 GHz ISM band, and utilizing hydrogen bonding to achieve the noninvasive detection of lactic acid (LA) in sweat by electropolymerization of dopamine with SiO_2_ nanoparticles to form a sensitive film. The sensor has a sensitivity of 150 KHz/mM, a detection range of 0–20 mM, omnidirectional radiation characteristics, biocompatibility, and dynamic stability, providing a new solution for the wireless monitoring of sweat metabolites [50].

In conclusion, the advantageous flexibility, lightweight quality, and compatibility of flexible substrates have led to the gradual replacement of traditional rigid substrates in wearable antenna designs. In order to optimize the performance and efficiency of wearable antennas in the design process, low-loss dielectrics are typically selected as the antenna substrate materials, while highly conductive materials are chosen for use as the antenna radiating elements. Table 2 provides a comprehensive list of the relevant parameters for some substrate materials.

### 3.2. Antenna Fabrication Techniques

Several primary techniques for designing and manufacturing on thin or flexible substrates include screen printing [52,53], sewing and embroidery [54,55,56], inkjet printing [57,58], and substrate-integrated waveguides (SIWs) [59,60]. These methods have been extensively utilized in industrial production. Choosing appropriate manufacturing technologies for various materials can enhance antenna performance while reducing manufacturing complexity and costs [5].

#### 3.2.1. Screen Printing

A screen with open areas is positioned on top of the substrate, through which the ink flows. During the printing process, the ink is compressed by a squeegee from the open areas of the screen onto the substrate, thereby forming the printed pattern. This method has the advantages of low cost and simplicity; however, it also has disadvantages. The prints produced have low resolution, and it is difficult to control the thickness of the conductive ink, which can result in a reduction in pattern resolution and antenna performance after multi-layer printing. Furthermore, this technique can only be used on specific substrates, and the depth of penetration of the conductive ink on the substrate must be considered, as this may impact the performance of the antenna.

As illustrated in Figure 4, Umar Hasni et al. [61] presented 5.8 GHz coplanar lock-hole fabric antennas and 915 MHz radio frequency identification (RFID) tag antennas for in vivo and ex vivo sensor communication, where a conductive ink is coated on a fabric substrate using a screen printing process, and a non-conductive coating is introduced as an interfacial layer to enhance the conductivity and fabric adhesion.

Hattan F. Abutarboush et al. employed the screen printing technique to deposit nanosilver ink on a flexible Kapton substrate. The resulting antenna exhibited a wide bandwidth of 1.77 GHz–6.95 GHz and a 119% bandwidth [62]. Chun Xu Mao et al. devised a screen-printed full-duplex textile antenna with enhanced isolation, operating in the ISM band at 2.45 GHz. This antenna features innovative integrated strips in both ports, which improve the bandwidth and isolation between the ports [63].

#### 3.2.2. Inkjet Printing

Conductive inks are deposited onto the flexible substrate at the desired location via a printer nozzle and then thermally cured to ensure that it is electrically conductive. The cost of the conductive ink, the printer, and the thermal curing technology contribute to the higher expense of this process compared to other manufacturing techniques. As illustrated in Figure 5a, the nanosilver ink is printed on PET paper using double-layer printing, followed by heat curing at 140 °C. In comparison to the simulated antenna, the resonance point of the second band of the fabricated antenna is shifted from 6.85 GHz to 7 GHz, with a peak gain in the range of 3 dBic–4 dBic [51].

Hang Yu designed a flexible coplanar waveguide antenna based on inkjet printing technology. The antenna is fabricated with polyimide substrate and silver nano-ink, the resonant frequency shift is ≤360 MHz, and the return loss is kept to within −14 dB at a lateral/longitudinal bending radius of >30 mm and a skin spacing of >1 mm, which is suitable both for anti-jamming capability and for human safety [64], as shown in Figure 5b. a compact tri-notched flexible ultra-wideband antenna based on inkjet printing and plasma-activated silver nano-inks is presented in [65], where the inks have good printing suitability and can produce highly conductive patterns on flexible PET substrates by efficient plasma sintering techniques.

Antennas fabricated through screen printing and inkjet printing exhibit poor washability due to the gradual degradation of the ink. To address this issue, conductive inks can be encapsulated with a thermoplastic coating for protection.

#### 3.2.3. Sewing or Embroidery

Conductive patterns are woven using traditional methods and then fixed to a substrate. Alternatively, conductive yarns can be woven onto a fabric substrate using an embroidery or sewing machine. In comparison to alternative antenna fabrication techniques, this approach enables the seamless integration of the antenna into clothing without compromising aesthetic appeal, while offering a reduced cost and a more straightforward fabrication process. The efficacy of embroidered antennas is contingent upon the tensile strength of the conductive yarn, the precision of the embroidery machine, and the density and direction of the embroidery stitches. Conversely, the conductive patterns produced by embroidery exhibit a relatively low resolution. Furthermore, the resolution of the geometric patterns is diminished when the material is subjected to bending and stretching. It has been demonstrated that the moisture and gum present in the fabric can be removed by stretching and boiling the fabric substrate prior to printing the ink. This process reduces the surface roughness of the fabric, thereby enhancing the adhesion of the ink to the fabric and improving the effective conductivity of the fabric antenna [66]. As illustrated in Figure 6a, the two dipole antennas designed by Sanjit Varma et al. used an embroidery machine to automatically sew the radiating structures with an accuracy of 0.1 mm. The embroidery process sews the copper conductive threads directly into the fabric, avoiding the constraints of a rigid substrate. Tests showed that the antennas maintain a return loss of <−14 dB when bent at a radius of 25.7 mm [67]. As illustrated in Figure 6b, Truong et al. [68] proposed a wearable RFID antenna that uses an embroidery machine to achieve <1 mm precision pattern molding, and adopts a single-sided embroidery process to reduce the risk of silver thread breakage due to friction during high-speed embroidery. A comparison of the single- and double-layer embroidery showed that the double-layer embroidery reduces surface discontinuities.

In comparison to screen-printed antennas, embroidered antennas demonstrate a higher measurement gain. Additionally, screen-printed antennas exhibit a notable impedance mismatch, whereas embroidered antennas are more resilient when assessing the performance variation of antennas at disparate locations on the human body [70]. As illustrated in Figure 6c, Sofia Bakogianni et al. proposed a fully textile reconfigurable dipole antenna based on an embroidery process using conductive nylon yarn and conductive Velcro to achieve mechanical frequency reconfiguration, and four states formed by folding the dipole arms to cover the 780–1330 MHz band (UHF/L band) [69].

#### 3.2.4. Substrate-Integrated Waveguides (SIW)

Substrate-Integrated Waveguide technology represents a cost-effective and relatively straightforward fabrication process. The structure combines the advantages of planar and non-planar guided structures by employing shorting vias in the cavity sidewalls, thereby ensuring that the electric field is confined within the cavity and supported by the entire grounding layer. The SIW technology can efficiently integrate a variety of passive components and systems onto a single substrate. The implementation of different SIW structures has the potential to facilitate miniaturization, to expand bandwidth, to enhance flexibility, and to improve gain [71]. As illustrated in Figure 7a, the circularly polarized antenna proposed in [72] uses the SIW technique to construct a low-profile cavity, radiates through the top annular slit, feeds through the microstrip transition, and introduces rectangular cutouts and short-circuited overbore probes to optimize the radiation performance. As illustrated in Figure 7b, Giovanni Andrea Casula et al. proposed a novel dual-band antenna, where the SIW technique reduced the antenna size to one-eighth of the square SIW [73]. As shown in Figure 7c, the antenna proposed in [74] uses an RT/duroid^®^ 5880 substrate with an L-shaped slot etched in the ground plane with shorting apertures introduced to achieve coverage in the 25.1–29 GHz band (3.9 GHz bandwidth, 14.4% relative bandwidth), with a peak gain of 7.5 dBi, and a size of only 14 × 18 × 0.787 mm^3^. Guo-Ping Gao et al. designed a tri-band wearable substrate-integrated waveguide textile antenna, as shown in Figure 7d. The antenna excites multi-frequency points through a SIW resonant cavity and introduces a hypersurface (MS) to regulate the frequency and bandwidth, which achieves three frequency bands coverage within a compact size and, at the same time, enhances the radiation efficiency by utilizing the hypersurface to achieve a peak gain of 9.27 dBi [75].

### 3.3. Antenna Miniaturization Technology

The objective of antenna miniaturization is to reduce the antenna’s size while maintaining the antenna’s resonant frequency. Alternatively, the goal is to reduce the antenna’s resonant frequency while maintaining the antenna’s size. Reducing the size of the antenna will inevitably affect its performance, including bandwidth and gain. Consequently, the current research focus on antenna miniaturization is to reduce the antenna’s size while maintaining the antenna’s other performance parameters.

#### 3.3.1. Dielectric Substrate with High Dielectric Constant

In accordance with the principles of cavity mode theory, the frequency of a rectangular microstrip antenna can be approximated by the following mathematical formula:(1)$$f=\frac{c}{2\sqrt{\varepsilon_{r}\mu_{r}}}\sqrt{{\left(\frac{m}{L}\right)}^{2}+{\left(\frac{n}{W}\right)}^{2}}$$

The dimensions “L” and “W” represent the length and width of the rectangular microstrip antenna, respectively. The symbols “µ_r_” and “ε_r_” denote the magnetic permeability and relative permittivity, respectively, of the dielectric substrate. Finally, “c” represents the speed of light in a vacuum, which is approximately 3 × 10^8^ m/s [76].

From the formula, it can be seen that the permittivity and magnetic permeability of the dielectric substrate are inversely proportional to the frequency of the antenna. Therefore, a material with high permittivity or high permeability can be selected in order to reduce the resonant frequency of the antenna, thus achieving the miniaturization of the antenna. Nevertheless, the utilization of a high-dielectric-constant dielectric substrate will result in the generation of a robust surface wave, which will consequently induce an increase in dielectric loss. This, in turn, will lead to a reduction in the radiation efficiency of the antenna. This phenomenon can be elucidated by the following equation:(2)$$Q_{min}\approx \frac{1}{{\left(k\alpha \right)}^{3}}+\frac{1}{k\alpha }$$

The quality factor, Q, is approximately equal to 1 divided by the bandwidth, BW, which is approximately equal to 1/Q. The antenna wave number, k, and the antenna size, α, are also relevant factors. Equation (2) demonstrates that, as the antenna size decreases, the bandwidth also decreases [76].

S. Sreelekshmi and S. Perumal Sankar proposed a portable circular patch antenna at 2.5 GHz using a substrate with a dielectric constant of 10.2. The overall dimensions of the antenna are 35 × 30 × 1.6 mm^3^, which can provide a gain of 2.5 dB with a radiation efficiency of 84% [77]. Marian G. Banciu et al. used a high dielectric constant material, alumina, as a substrate to design the microstrip antenna and printed monopole antenna in the 5.8 GHz band. The microstrip antenna provides a narrow bandwidth of 30 MHz and the monopole antenna has a wide impedance bandwidth between 3.6 GHz and 6.35 GHz [78].

#### 3.3.2. Surface Grooving

An increase in the current path length can be achieved by modifying the shape of the patch antenna or by creating slots on the antenna surface. This results in an equivalent increase in the effective length of the antenna. However, this approach also generates additional currents on the antenna surface, which leads to an increase in cross-polarization and a subsequent reduction in the gain and efficiency of the patch antenna. As in Figure 8a, the resonant frequency of the antenna can be reduced by loading a U-shaped slot on the patch antenna to achieve the desired operating frequency. Finally, a slot is added at the upper right corner to enhance the control of the current distribution [79]. As illustrated in Figure 8b, the resonant frequency can also be regulated by incorporating two L-shaped slots into the patch antenna [80]. In Figure 8c, Kummaramsetty Sainath et al. devised a multi-slot antenna through the etching of slots, which served to extract the radiation pattern from the conventional rectangular patch shape. This approach was undertaken with the objective of reducing the antenna’s operating frequency. Additionally, rectangular slots were etched at the ground plane on the back of the substrate, with the aim of improving impedance matching at the resonance frequency [81]. Jeena George et al. proposed an elliptical slotted patch antenna to redirect the frequency of the antenna to the desired band range by etching varying numbers of elliptical slots on a conventional rectangular patch, as shown in Figure 8d [82].

#### 3.3.3. Short-Circuit Pins

The introduction of shorting probes on the microstrip antenna, situated in close proximity to the feed point, resulted in the formation of a coupling capacitance, thereby facilitating the miniaturization of the antenna. In a similar approach, Farzad Khajeh-Khalili et al. employed a comparable methodology by loading the antenna with three pairs of shorting pins, thereby effectively eliminating the excess resonant frequency and generating a secondary band (2400–2480 MHz) without altering the antenna’s dimensions [80]. Similarly, Bin Hu et al. employed the use of shorting columns on the antenna, achieving a similar outcome. The proposed configuration maintains superior performance even when subjected to bending, as evidenced by the transition from an F-shaped radiating patch to an E-shaped configuration. This modification has the additional benefit of reducing the antenna size from 96 × 47 mm^2^ to 70 × 25 mm^2^. Furthermore, the minimum value of S11 measured in space is reduced from −14.59 dB to −33.30 dB, which effectively realizes the miniaturization of the antenna [83].

### 3.4. Frequency Selective Surface (FSS)

Frequency selective surfaces (FSSs) represent an important class of artificial electromagnetic structures, consisting of periodic metallic patches or apertures that act as spatial filters for electromagnetic waves. By appropriately designing the unit-cell geometry, FSSs can selectively transmit or reflect specific frequency bands, thereby enabling applications in antennas, radomes, and stealth technology. Building on the concept of FSSs, artificial magnetic conductors (AMCs) were developed as surfaces that exhibit a reflection phase close to 0°, unlike conventional metals which reflect with a 180° phase. AMC surfaces can enhance antenna performance, suppress surface waves, and reduce antenna profile, and they are often realized using periodic FSS structures.

Metamaterials can be classified into two principal categories: three-dimensional metamaterials and two-dimensional hypersurfaces. Metamaterials were initially proposed in 1968 by Prof. Viktor G. Veselago in a paper that introduced the concepts of left-handed materials and double-negative materials, which are materials with negative dielectric constant and magnetic permeability [84]. Subsequently, Sir John B. Pendry et al. experimentally verified the existence of a negative dielectric constant and negative permeability [85,86]. In 2014, the Chinese Academy of Sciences academician Tie Jun Cui et al. proposed the concepts of coded metamaterials, digital metamaterials, and programmable metamaterials. By designing 1-bit/2-bit phase discretization units and coding sequences, electromagnetic wave manipulation and RCS (Radar Cross-Section) reduction were achieved [87].

An electromagnetic bandgap (EBG) is a collective term for periodic structures that can propagate or suppress electromagnetic waves in specific frequency bands. Attaching an EBG structure to the backside of an antenna radiating patch can suppress the surface waves of the antenna, thus realizing the purpose of antenna miniaturization. In 1999, Dan Sievenpiper et al. proposed the classical EBG structure, which is mushroom-like in shape [88].

In [89], a microstrip patch antenna sensor based on an artificial magnetic conductor was proposed for the noninvasive early diagnosis of breast and brain cancer, as shown in Figure 9a. The antenna introduces an additional resonance at 2.276 GHz through the artificial magnetic conductor (AMC) layer, and the peak gain is boosted to 11.7 dBi. Tumors with a minimum radius of 0.5 mm are successfully detected in breast and brain tumor models, demonstrating high sensitivity. As illustrated in Figure 9b, Ashfaq Ahmad et al. devised two wearable antennas equipped with distinct EBG crystalline cell arrays. The dual-band EBG microstrip antenna exhibited a notable enhancement in directionality, gain, efficiency, and other characteristics compared to the conventional antenna devoid of an EBG-loaded structure. The two antennas demonstrated consistent resonance during bendability tests [90].

Usman Ali et al. proposed a dual-frequency flexible wearable antenna, as depicted in Figure 9c, which is well-suited for 2.45 GHz and 5.8 GHz wireless body area network (WBAN) applications. By incorporating an AMC array, the antenna’s performance was significantly enhanced, resulting in a 98% reduction in the specific absorption rate (SAR) for both 1 g and 10 g of human tissue [91]. As illustrated in Figure 9d, Fatima-ezzahra Zerrad et al. developed a slot ultra-wideband antenna featuring an AMC structure. This design boasts a bandwidth of 7 GHz, along with achieving a radiation efficiency exceeding 85% and a realized gain of 5.1 dBi [92].

### 3.5. Antenna Feeding Technology

Two principal categories of microstrip antenna feeds have been distinguished, contact feed and non-contact feed. Coaxial feed and microstrip line feed are classified as contact feeds, while neighbor coupling feed and aperture coupling feed are designated as non-contact feeds.

#### 3.5.1. Coaxial Feed

Coaxial feed technology establishes a connection between the outer conductor of the coaxial probe and the ground layer, while the inner conductor penetrates the dielectric and establishes contact with the patch. The coaxial probe introduces an inductance that is dependent upon its length and is determined by the thickness of the antenna’s substrate, in addition to the radiation generated by the probe. The primary advantage of this technique is that the coaxial feed can be placed at any point within the patch to match the input impedance. However, this technique is not suitable for textile applications where comfort and flexibility are important considerations.

#### 3.5.2. Microstrip Feeding

In this feed method, microstrip lines are etched on the surface of the substrate to ensure the planar structure of the antenna, thereby eliminating the need for a coaxial probe to pass through. This reduces the difficulty of antenna fabrication and increases antenna flexibility. Impedance matching is achieved by modifying the length and width of the microstrip line, which increases the size of the antenna and the complexity of impedance matching. Deisy Formiga Mamedes et al. compared four impedance matching techniques based on microstrip line feeds. These include embedded feeds, offset feeds, quarter wavelength impedance converters, and mixed-method feeds, which provide larger bandwidths and higher broadside gains [93].

#### 3.5.3. Coplanar Waveguide Feed (CPW)

CPW technology is based on microstrip feed technology, which is a planar structure comprising a center conductor and two adjacent ground planes. This technology offers several advantages, including low loss, wide bandwidth, and ease of integration. The impedance matching of CPW technology is achieved through modifications to the width of the microstrip line and the gap between the microstrip line and the ground plane. In Figure 10a, Adel Y. I. Ashyap et al. proposed a 2.4 GHz wearable CPW antenna to enhance its performance by incorporating an EBG-FSS structure, which led to a 13 dB improvement in the front-to-back ratio (FBR), a 6.55 dBi increase in gain and, in comparison to the antenna alone, a reduction of the SAR greater than 95% [94]. An asymmetric ACS-fed monopole antenna, as illustrated in Figure 10b, was designed by Ansal Kalikuzhackal Abbas and Shanmuganatham Thangavelu. The performance of the antenna was enhanced by transitioning from a basic CPW feed to an ACS feed [95].

#### 3.5.4. Adjacent-Coupled Feed

The adjacent-coupled feed approach involves the placement of the feed line between two dielectric substrates, with the radiating patch situated on the uppermost substrate, as illustrated in Figure 11 [96]. The neighborhood coupled feeding technique has been demonstrated to produce good cross polarization in comparison to other feeding methods. The use of a fully grounded planar structure and a thick substrate has been shown to increase body isolation and to reduce the absorption of electromagnetic radiation by human tissues. Furthermore, the characteristics of high bandwidth and unidirectional radiation direction map align with the design requirements of wearable antennas [97].

### 3.6. Bionic Design of Wearable Antennas

Bionic antennas with wider bandwidth and smaller size have become an alternative to conventional antennas. From the biologically inspired butterfly-like antenna [98] to the leaf antenna [99] and the snowflake antenna [100] in the abiotic world, bionic structures provide antenna engineers with more design ideas. As shown in Figure 12a, the orchid-shaped antenna proposed by [101], based on the perturbation method (PM), modifies the shape of the traditional circular patch antenna and achieves miniaturization by introducing symmetrical petal-like edges, which maintains high gain over a wide bandwidth compared to the traditional design. Gangadhara Mishra and Sudhakar Sahu designed a dual-band tree-shaped microstrip patch antenna, as shown in Figure 12b, with ultra-broadband characteristics from 2.2 GHz to 19.5 GHz in the IEEE802.16 standard (Wi-MAX) band [102] and wireless local area network (WLAN) band resonance, and achieved the highest gain of 7.5 dB over the entire operating band [103]. As shown in Figure 12c, Asutosh Mohanty and Sudhakar Sahu designed a four-port multiple-input–multiple-output (MIMO) antenna based on the shape of maple leaf pods with high bandwidth, omnidirectional radiation pattern, and an SIW structure that can provide higher body isolation [104].

## 4. Performance of Wearable Antennas

### 4.1. Specific Absorption Rate

Wearable antennas must meet safety standards because they produce back radiation, which poses a health risk to humans. Internationally, the SAR is used to measure the effect of electromagnetic radiation on the human body. The specific absorption rate (SAR) of an antenna is the amount of energy absorbed by the human body per unit of mass, and is expressed in units of W/kg. It is calculated by the following formula:(3)$$SAR=\frac{\sigma {\left|E\right|}^{2}}{\rho }$$
where σ is the tissue conductivity unit in S/m, E is the electric field unit in V/m, ρ is the tissue mass density unit in kg/m^3^. At present, the world SAR standard value mainly uses the International Commission on Non-Ionizing Radiation Protection (ICNIRP) standard, 10 g of tissue SAR value should be less than 2 W/kg [105], as well as the International Institute of Electrical and Electronics Engineers (IEEE) standard, the SAR value of 1 g tissue should be less than 1.6 W/kg [106], while China has adopted the GB 8702-2014 “Electromagnetic Environment Control Limits” issued by the Ministry of Environmental Protection in 2014.

Mahmoud A. Abdelghany et al. [107] proposed a flexible, wearable dual-band monopole antenna fabricated on denim textile, covering two wide bands of 2.2–4.0 GHz and 5.0–10.0 GHz. The measured SAR values at the left head were 1.09, 0.92, and 1.08 W/kg at 2.45, 3.50, and 5.80 GHz, respectively. For the stomach, the corresponding SAR values were 0.53, 0.73, and 0.44 W/kg, and for the right head they were 0.74, 0.97, and 1.13 W/kg. These results indicate that the antenna complies with the relevant international safety limits. Tale Saeidi et al. proposed a dual-band MIMO wearable antenna fabricated from conductive textile materials on a textile substrate, with approximate dimensions of 0.37λ_0_ × 0.25λ_0_ × 0.01λ_0_. The antenna operates in two bands around 3.25–3.65 GHz and 5.4–5.85 GHz. The SAR simulations were performed at 3.4 GHz and 5.6 GHz and reported as SAR averaged over 1 g and 10 g of tissue; under typical on-body configurations the local SAR values are well below the commonly accepted safety limits. However, the local peak SAR values of ≈1.44 W/kg (tissue model) and ≈2.6 W/kg (voxel model) were observed for close placements (antenna-to-tissue distance ≈ 5–8 mm). Overall, conventional on-body simulations indicate compliance, but the elevated peaks at extreme proximity warrant attention in safety assessments [108].

In the experiment, the SAR distribution is governed by multiple coupling factors, including transmission power, the distance and relative position between the antenna and the skin, operating frequency (which determines the penetration depth and energy deposition characteristics of electromagnetic waves), antenna pattern and polarization state, antenna structure and its shielding design, scattering effects from clothing or nearby metallic objects, and the geometric shape and tissue layering of the test subject (e.g., skin, fat, muscle, etc., which have different dielectric parameters). In numerical simulations and experimental measurements, the peak SAR is highly sensitive to spatial average mass (commonly 1 g or 10 g) and grid/voxel resolution. Common engineering measures to suppress the local SAR include reducing the transmission power, increasing the distance between the antenna and the skin, using directional or grounded antenna designs to direct energy away from the human body, and dispersing energy through antenna layout and system-level optimization to avoid local energy concentration.

### 4.2. Bendability

Wearable antennas are designed to be integrated with clothing and must be able to conform to the natural curvature of the human body. This includes the arms, legs, face, and chest, as well as the human body in motion. The antenna’s ability to maintain its shape while adapting to these variations is crucial for its functionality. Therefore, understanding the antenna’s curvature performance is a vital aspect of its development.

As illustrated in Figure 13a, T. Aishwarya and Priyanka Das designed a wearable modified bow-tie antenna with an integrated frequency selective surface (FSS) for biomedical applications. The gain decreases slightly from 7.6 dBi to 7.2 dBi, and the radiation pattern remains stable, which indicates that the design is well adapted to the human body surface [109]. As illustrated in Figure 13b, the wearable dual-band antenna devised by Umar Musa et al. operates at 2.4 GHz and 5.8 GHz. The impact of bending the antenna along the *x*-axis and *y*-axis on its performance was investigated, and the findings indicate that bending enhances the backward radiation of the antenna, resulting in a shift in frequency [110].

Deepti Sharma et al. proposed a multi-band wearable textile antenna that employs a stepped square monopole antenna structure with an asymmetrical curved slot at the top and a partially slotted ground plane at the bottom. The antenna has impedance bandwidths of 310 MHz, 960 MHz, and 1140 MHz at 1.8 GHz, 2.45 GHz, and 5.8 GHz, respectively, with gains of 3.7 dBi, 5.3 dBi, and 9.6 dBi, respectively. Studies indicated that even under different bending radii (e.g., 25 mm, 35 mm, and 45 mm), the antenna’s reflection coefficient and frequency response exhibit minimal variation, demonstrating its excellent flexibility and adaptability [111]. Rakesh N. Tiwari et al. [112] proposed a four-band 1 × 4 linear MIMO array antenna. The antenna measures 11.0 × 44.0 mm^2^ and is printed on a flexible substrate, supporting four frequency bands: 26.28–27.36 GHz, 27.94–28.62 GHz, 32.33–33.08 GHz, and 37.59–39.47 GHz. The antenna achieves peak gains of 6.12 dBi, 8.06 dBi, 5.58 dBi, and 8.58 dBi at its four resonance frequencies, with a total radiation efficiency exceeding 75%. Measurement results indicate that the antenna maintains stable performance under various bending conditions (Rx = 40 mm, 60 mm, 80 mm; Ry = 25 mm, 50 mm, 75 mm).

Table 3 summarizes the antenna parameters and bending radii during bending performance testing from selected references.

## 5. Applications of Wearable Antennas

The applications of wearable antennas encompass a broad spectrum of wireless communication and sensing needs, spanning a range of frequencies from low to high, contingent upon the specific frequency band. Each frequency band exhibits distinct requirements for antenna design, performance, and application scenarios.

Within the low-frequency band (30 Hz–3 MHz), wearable antennas find application in the wireless transmission of medical devices, particularly those used for the monitoring of electrophysiological signals, such as electrocardiograms (ECGs) and electroencephalograms (EEGs). The penetration capability of low-frequency signals, which is attributed to their high frequency, enables effective transmission through the human body. In the UHF band (300 MHz–3 GHz), wearable antennas facilitate wireless communication between wearable devices, including heart rate monitors, blood glucose monitors, health bracelets, and smart clothing. In higher microwave bands (3 GHz–30 GHz) and millimeter-wave bands (30 GHz–300 GHz), the focus has shifted to real-time transmission of large data sets, such as high-definition video, and real-time monitoring and control of internet of things (IoT) devices. The subsequent discussion will primarily address the applications of wearable antennas in the microwave band.

### 5.1. Cancer Detection

Wearable antennas can be used to monitor patient’s vital signs, including body data such as heartbeat, temperature and blood pressure, and can also be used for the detection of some diseases [113], including the detection of cancers and tumors such as breast cancer [114], skin cancer [115], kidney cancer [116], and thyroid cancer [117,118].

Komalpreet Kaur and Amanpreet Kaur proposed an ex vivo detection method for skin cancer based on ultra-wideband stacked microstrip patch antennas. The proposed antenna structure is compact and has a peak gain of 6.3 dB at a frequency of 10.8 GHz [119]. Raja Rashidul Hasan et al. developed a patch antenna based on multi-walled carbon nanotubes which can be used to aid in the detection of lungs affected by SARS-CoV-2. The antenna demonstrated favorable performance in the frequency band of 6.63–7.29 GHz, exhibiting optimal characteristics within this range [120]. Demyana A. Saleeb et al. designed a reconfigurable circularly polarized antenna array for the detection of brain cancer, as illustrated in Figure 14a. The antenna was validated through simulation experiments, demonstrating the capability to detect tumors of 5 mm and 2.5 mm in size [121]. The planar antenna array proposed in [122] is used for tumor detection in microwave breast imaging systems. The antenna consists of a square concentric ring, which improves the average radiation gain by 4.8 dBi and the efficiency by 18% compared with the traditional square patch antenna array. Experimental validation by simulating a human breast model shows that the imaging resolution and contrast are better than those of the conventional antenna in detecting a 5 mm tumor, as illustrated in Figure 14b.

The technique of imaging by microwave was initially employed for the detection of breast cancer. Despite the complexity of the dielectric properties of human tissues and the dearth of pertinent in vivo measurement data, researchers have devised a range of clinical or simulation experiments to conduct the requisite measurements [123]. Manisha Ghosh and Banani Basu designed a UWB monopole patch antenna for the microwave imaging of malignant breast tissues, with the objective of discriminating between benign and malignant tissues based on the SAR variation curves at a specific resonance frequency. This approach is founded on the premise that the dielectric constant of tumors is higher than that of normal tissues. When combined with a deep learning model, the accuracy of discriminating malignant tissues reached 98.59% [124].

As illustrated in Figure 15, S. Sadasivam and Thulasi Bai devised a UWB antenna for breast cancer detection, which operates within the 3 GHz–10.2 GHz band and achieves 92% efficiency and 3.2 dBi gain across the entire band, thus satisfying the internationally mandated electromagnetic exposure limits [125].

Md Samsuzzaman et al. presented a circular slotted defect grounded monopole patch antenna for microwave head imaging to detect brain tumors, as shown in Figure 16. The antenna is based on an FR4 substrate and achieves an ultra-wide bandwidth of 1.22–3.45 GHz with a gain of more than 5 dBi and a radiation efficiency of >85% through the design of a circular slot in the center, a triangular tangent angle, and a partially grounded planar surface. In Hugo model validation, the antenna is able to effectively identify 5 mm tumors with 30% bandwidth and 2 dBi gain improvement over conventional antennas [126].

### 5.2. Vital Signs Monitoring

In the field of sports, wearable antennas have the potential to be utilized for the non-invasive monitoring of an athlete’s health status and athletic performance.

As illustrated in Figure 17, Mariam El Gharbi and colleagues devised an embroidered antenna that can be incorporated into commercially available T-shirts for real-time respiration monitoring through a technique based on chest motion analysis. Monitoring is based on the resonance frequency shift of the zigzag dipole antenna sensor, which is induced by the expansion of the chest and the displacement of the air volume in the lungs during breathing. The proposed wearable antenna has been demonstrated in an experimental setting to detect respiration with guaranteed accuracy in two different postures (standing, sitting) [127]. Similarly, Mehran Ahadi et al. proposed a half-wave dipole antenna [128] that can be used for strain sensing applications, such as respiration detection. The antenna is composed of a conductive polymer and has been tested for its durability and flexibility. The antenna exhibits a sensitivity that is 5.5 times greater than that of another antenna sensor for respiration detection [129].

In a separate study, a patch antenna incorporating a silver mode was employed to quantify oxygen concentration. The resonant frequency of the antenna exhibited a corresponding change in accordance with the oxygen concentration. The designed antenna has the potential for integration into healthcare systems [130].

As illustrated in Figure 18, Jessica Hanna and colleagues devised a multi-sensor system comprising a multi-frequency slit antenna and a multi-frequency rejection filter. This system can be employed to monitor fluctuations in blood glucose levels in diabetic patients in real time by integrating it into wearable gloves. Following validation in animal experiments and human trials, the proposed wearable sensor system has been shown to have significant advantages. [131].

The advancement of antenna technology has also led to the development of microwave-based solutions for the detection of potential bone-related issues. In response to the rising prevalence of osteoporosis, Dr. Kiran Rathod et al. proposed a hexagonal loop wearable antenna. The antenna employs polyester conductive e-textile as the radiating fabric and polyester fabric as the substrate, operating within the frequency range of 1.5 GHz–4.5 GHz. Its efficacy for bone detection was validated through experimental verification [132].

Sumon Modak et al. [133] reported an ultra-wideband antenna that is boasting and miniaturized, exhibits better gain, compactness and robustness in measurements, and adapts to the curvature changes of the human body, which can be applied to human health monitoring in both indoor and outdoor environments. Pei-Yu He’s team designed a monopole antenna loaded with an AMC structure, which, in comparison with a conventional cardiopulmonary stethoscope (CPS) antenna, the designed antenna has better performance and higher sensitivity [134]. The antenna reported in [135] has good scalability, can be integrated into commercially available T-shirts, and can continuously monitor different human breathing patterns.

### 5.3. Other Applications

Wearable antennas have the potential to be utilized in military settings for the purpose of personnel localization. Additionally, they could prove invaluable in emergency situations, such as disasters or accidents, for the localization of injured personnel and the facilitation of emergency rescue operations.

Rishabh Kumar Baudh et al. designed a circularly polarized fabric antenna for defense applications, which is loaded with a hypersurface structure to achieve a wide bandwidth of 37.5% (7.73 GHz–11.3 GHz), a high gain of 8.9 dBic, and an axial-ratio bandwidth of 3 dB of 21.3% (8.8 GHz–10.9 GHz) [136]. Esra Çelenk and Nurhan Türker Tokan devised a badge antenna for military applications. The antenna operated at 8 GHz with a gain and efficiency of 5.2 dBi and 79.2%, respectively. Additionally, the peak specific absorptions were measured to be 0.53 W/kg and 0.69 W/kg in male and female chests, respectively [137]. A wearable antenna was designed by Alicia Flóres Berdasco et al. [138] for electronic travel assistance applications, which is worn on the user’s arm and uses the body’s movement to help the user walk by imaging and detecting obstacles through radar technology.

Table 4 Summarizes the basic parameters of antennas for wearable antenna applications.

## 6. Perspectives of Wearable Antennas

As communication technology continues to advance, so too do the demands placed on wearable antennas [139]. The integration of communication systems into people’s daily lives is a potential future direction, and it is therefore essential that wearable antennas are capable of meeting the requirements of this new era. The objective is to develop more sophisticated wearable antenna systems that provide enhanced services to users by integrating multiple antennas and to design high-performance wearable antenna systems. There are numerous advanced methodologies, including massive MIMO, antenna arrays, metamaterials, artificial magnetic conductors, and dielectric superlayers. Another avenue of research is the miniaturization of wearable antennas, which reduces their size and facilitates integration into everyday clothing and smaller accessories. This design approach enhances user comfort and convenience.

In the field of healthcare, implantable antennas offer a promising advancement in wearable healthcare technology. However, their design faces challenges due to the complex dielectric properties of human tissues, making a theoretical analysis of radiation characteristics difficult. Most studies rely on simplified human body models, limiting accuracy. Despite these challenges, implantable antennas hold great potential for long-term health monitoring and early disease detection.

In terms of frequency band, the 6G terahertz band is situated between the microwave and far-infrared spectra. It produces non-ionizing radiation that causes minimal or no harm to the human organism. Additionally, the long terahertz wavelengths can penetrate certain non-metallic materials, making terahertz antennas suitable for medical imaging, high-speed communication, and other applications. At high frequencies (5G millimeter wave, 6G terahertz), the antenna has a small structure, which presents a significant challenge for fabrication using traditional PCB technology. However, additive manufacturing technology offers a promising alternative in terms of cost-effectiveness and ease of fabrication [140].

With regard to materials, wearable antennas are required to function in close proximity to the human body over extended periods of time. This necessitates the consideration of materials suitable for use in sensitive areas, such as the eyes and the head. The selection of biocompatible materials can prevent the production of inflammation and other adverse reactions, as exemplified by silicon carbide [141]. Conversely, in order to ensure the long-term wearability of the device without compromising the performance of the antenna, it is essential to select flexible materials that can adapt to changes in human movement.

From the design side, the most commonly used computational electromagnetic (CEM) methods for antenna design are numerical methods and radio frequency methods. Three numerical analysis methods are commonly employed for the design and testing of antennas: the finite-difference time-domain (FDTD) method, the finite-element method (FEM), and the method of moments (MoM). The simulation of antennas necessitates the establishment of boundary conditions in computers to facilitate the resolution of partial differential equations. Commercially available software for antenna design includes high frequency structure simulators (HFSS), advanced design systems (ADS), and computational simulation techniques (CST). However, the execution time of HFSS and CST is considerable, increasing with the size of the antenna structure. Furthermore, ADS is unable to model the three-dimensional structure in detail. The advent of artificial intelligence (AI) has led to the integration of machine learning (ML) and deep learning (DL) into antenna design and optimization. Furthermore, AI techniques have the potential to reduce computational costs, to accelerate design processes, and to enhance the efficacy of antenna optimization [142].

Aggraj Gupta et al. devised a tandem neural network architecture that enables the fabrication of single- and dual-band antennas with more compact dimensions and reduced substrate thicknesses compared to conventional antennas, while maintaining competitive performance [143]. Duo-Long Wu et al. put forth a deep learning (DL) algorithm based on the backpropagation algorithm. This algorithm defines the inputs as the desired objectives of the two-port impedance bandwidth, average isolation, and maximum gain. By contrast, the outputs are defined as the antenna dimensions. Furthermore, a high isolation of greater than 40 dB is achieved over an operating bandwidth of 3.47 GHz to 3.58 GHz with a maximum gain of 4.3 dBi for both ports. The experimental results demonstrated that the predicted values of bandwidth and the actual gain values obtained by the MIMO-BP method are in perfect agreement with those reported in reference [144].

Arpan H. Shah and Piyush N. Patel designed a wearable textile antenna for use in diagnostic applications related to knee effusion and optimized the antenna’s parameters using a machine learning algorithm, resulting in improved impedance matching. The proposed DNN model predicted an optimal impedance matching of −36.21 dB, which is in close agreement with the −35.69 dB calculated by the CST simulation. The proposed ML algorithm is accurate and efficient [145]. Ahmed M. Montaser and Korany R. Mahmoud obtained 150 data samples to train the DNN model by changing the antenna geometry and related parameters, which is suitable for resonant frequency prediction [146].

As shown in Figure 19, S. Venkat et al. proposed a tri-band circularly polarized hexagonal patch antenna design method for 5G, combining the optimization of the Siamese Heterogeneous Convolutional Neural Network (SHCNN) with the Circle-Inspired Optimization Algorithm (CIOA) to extract antenna geometric and performance features through SHCNN, and to optimize the design parameters through CIOA to achieve coverage of different frequency bands. In comparison with the conventional algorithm, this algorithm reduces the number of iterations and shortens the design time by approximately 40% [147]. In [148], a miniaturized coplanar waveguide (CPW)-fed ultra-wideband (UWB) circularly polarized antenna was proposed, and the research team utilized a three-layer neural network to simultaneously optimize the optimal reflectance coefficient and the axial ratio for the first time, which reduces prediction errors and achieves accurate modeling of complex nonlinear relationships compared to traditional algorithms. An improved honey badger algorithm (GST-HBA) based on tent chaos and golden sine mechanism was proposed in [149].

GST-HBA significantly outperforms HBA, differential evolution, JAYA and sine cosine algorithm (SCA) in convergence speed, optimization accuracy and global searching ability, and provides a new method for solving complex electromagnetic problems. Wendong Yang et al. proposed an antenna design method based on the optimized agent model and the NSGA-III algorithm, which introduces the TCN network into the antenna agent model and solves the problem of balancing the efficiency and accuracy of the traditional CNN/BiLSTM in parameter mapping, and further reduces the running time by 2.67%, improves the prediction accuracy by 12.489%, and improves the computational efficiency significantly compared with the traditional scheme. The efficiency is significantly improved [150]. Table 5 summarizes the literature related to antenna optimization.

As a fundamental element of wearable devices, the design and fabrication of antennas directly impact the performance, safety, and user experience of these devices. The advent of flexible and novel conductive materials has prompted a shift in the design of wearable antennas from rigid to lightweight, flexible, and multifunctional, enhancing both comfort and aesthetics while ensuring efficient wireless communication. Research has demonstrated that the selection of conductive and substrate materials is pivotal in determining the antennas’ conductivity, dielectric properties, and compatibility. The advent of advanced manufacturing technologies, such as inkjet printing and screen printing, has further propelled the production of high-performance flexible antennas, thereby facilitating the proliferation of wearable devices. The imminent advent of 6G communication, millimeter wave, and terahertz technology is poised to further enhance the capabilities of wearable antennas, particularly in high-frequency applications. These high-frequency band antennas boast a wide spectrum, high data transmission rate, and strong resolution, and will be widely used in ultra-high-speed wireless communication, precise medical monitoring, augmented reality, and virtual reality devices, among other fields. In the medical field, high frequency antennas can realize the high precision monitoring and real-time data transmission of human biological signals, providing technical support for remote diagnosis and treatment and health management. In the domain of consumer electronics, these antennas will facilitate the enhancement of computing and communication capabilities in smart wearable devices, ensuring seamless connectivity.

In the future, the combination of flexible materials, nanotechnology, and artificial intelligence will promote the miniaturization, low power consumption, and high gain of antennas. This combination will support multi-frequency band adaptive functions and further enhance device performance and user experience. Concurrently, these technologies will also create new opportunities in the fields of health monitoring, smart apparel, and security monitoring. These opportunities will become an important cornerstone for the development of future communication technologies.

## 7. Comparisons

In recent years, a growing body of review papers has emerged on the subject of wearable antennas. The work of Marterer et al. delved into the development of wearable textile antennas, with a particular focus on material selection, fabrication methods, design strategies, and application areas [46]. Ref. [151], an article in the same field, paid close attention to the effect of different positions of the antenna in the garment—especially the distance between the skin and the fabric substrate—on its performance. Chishti et al. [113] discussed a variety of biomedical antennas for disease detection, analyzing several biocompatible materials as well as the bending properties of the antennas. Another paper focused on the properties of flexible substrate materials used for wearable antennas and the bending properties of antennas made of some polymer materials [25]. Singh et al. [152] systematically analyzed the design, structure, and application of flexible wearable antennas, focusing on their role in wireless body area networks (WBAN) and medical health technology, emphasizing the importance of maintaining optimal performance under conditions of human body contact, deformation adaptation, and environmental stress. Yang et al. [153] discussed the new materials used in wearable electronic devices. These studies have played an instrumental role in the design and development of wearable antennas. However, they are not without limitations. This review further strengthens the systematic nature of research on wearable antennas by incorporating the most recent studies in the field from the last five years. It clarifies the connections between materials, manufacturing processes, advanced design techniques, and application scenarios, while focusing on the integration of artificial intelligence with antenna design.

## 8. Conclusions

This review provides a systematic and comprehensive overview of the design of wearable antennas based on the latest 5 years of the literature. This article highlights the rapid development of wearable technology, driven by advances in wireless communication technology and the growing demand for wearable devices in various fields, such as healthcare, sports, and defense. This article pays particular attention to flexible substrates, novel materials (e.g., conductive fabrics), emerging fabrication methods (e.g., inkjet printing and embroidery), and the application of wearable antennas for health detection, and discusses the use of artificial intelligence in antenna design. Overall, it provides valuable guidance for future research and practical applications in the field of wearable antennas

## Figures and Tables

**Figure 1 micromachines-16-01028-f001:**
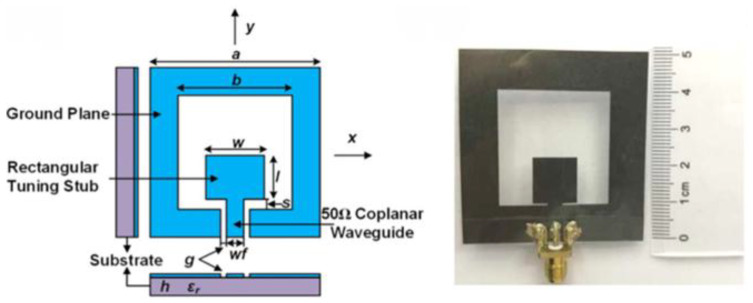
GAF wearable antenna [26].

**Figure 2 micromachines-16-01028-f002:**
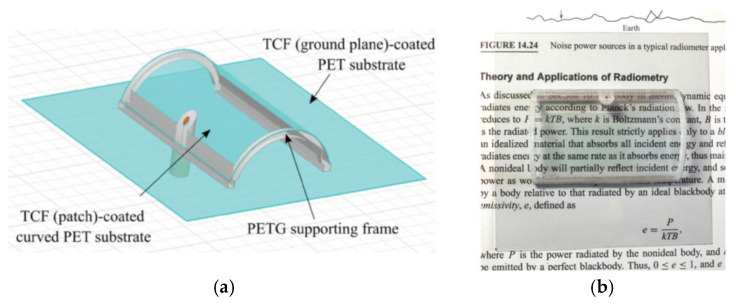
(**a**) Three-dimensional view of the proposed TCF curved patch antenna. (**b**) Photo of the manufactured prototype [28].

**Figure 3 micromachines-16-01028-f003:**
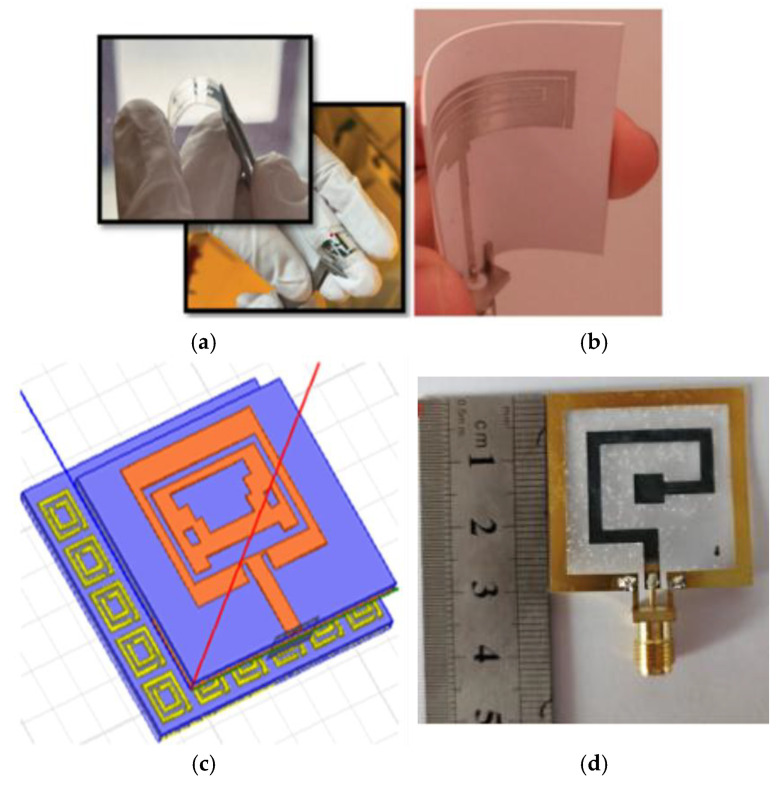
(**a**) Planar inverted-F antenna on chitosan substrate [47]. (**b**) Flexible antenna on synthetic paper substrate [48]. (**c**) Dual-band antenna loaded with metamaterials [49]. (**d**) The assembled antenna [50].

**Figure 4 micromachines-16-01028-f004:**
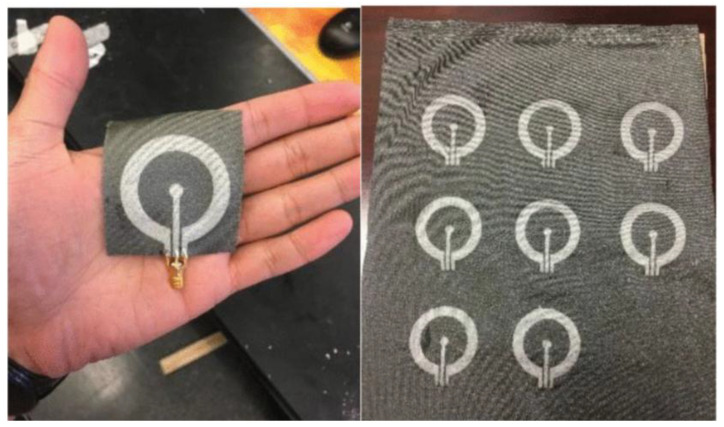
Screen printed CKA on a fabric (nylon/spandex) substrate [61].

**Figure 5 micromachines-16-01028-f005:**
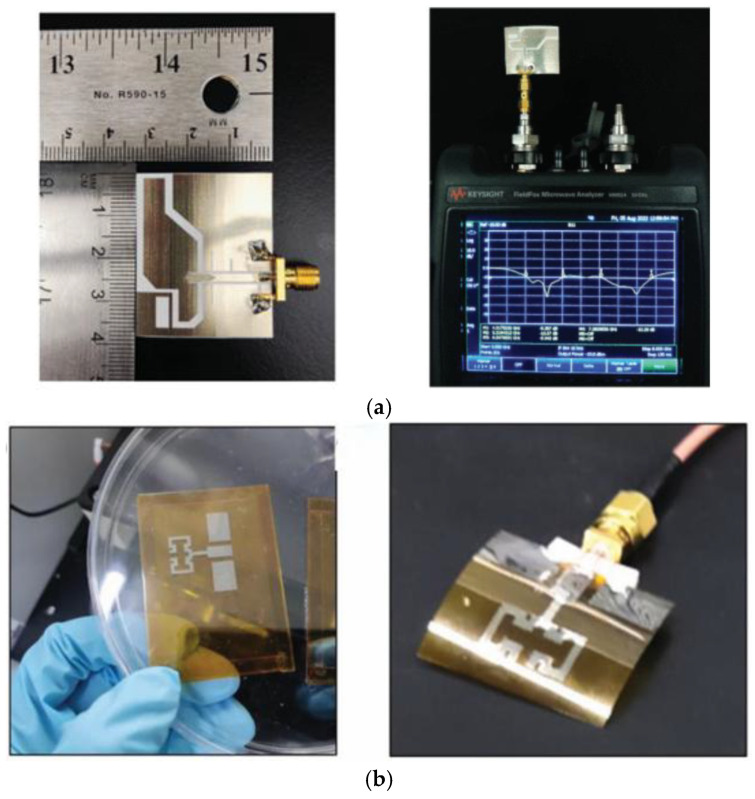
(**a**) Inkjet-printed antenna and antenna measurement [51]. (**b**) Antenna based on inkjet printing [64].

**Figure 6 micromachines-16-01028-f006:**
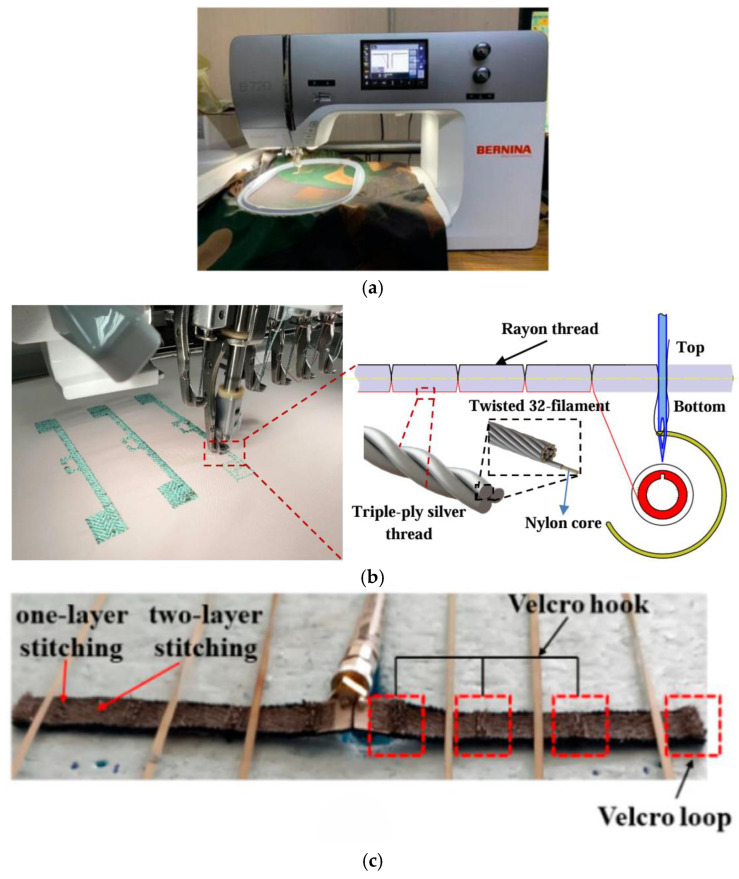
(**a**) The Bernina 720 sewing machine tool used for sewing the radiating element structures in the form of embroidered patterns over the denim-cotton fabric [67]. (**b**) Embroidery process [68]. (**c**) Fully textile reconfigurable dipole [69].

**Figure 7 micromachines-16-01028-f007:**
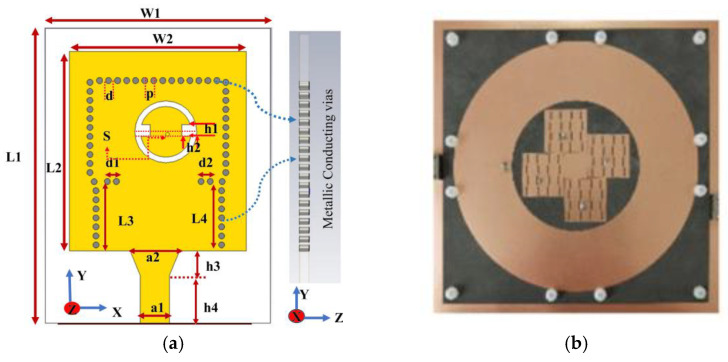
(**a**) Proposed SIW cavity backed antenna configuration with parameters [72]. (**b**) Dual-band wearable eighth-mode SIW antenna [73]. (**c**) Top view and bottom view of the proposed antenna [74]. (**d**) Configuration of the proposed antenna [75].

**Figure 8 micromachines-16-01028-f008:**
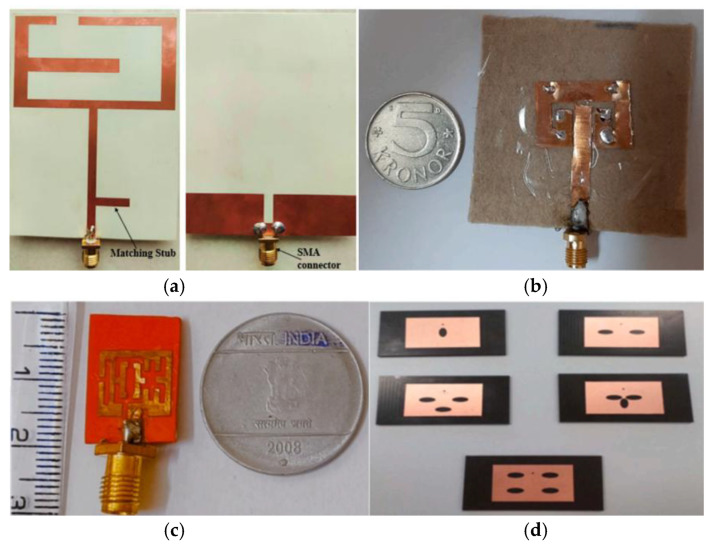
(**a**) Wearable remote (LoRa) patch antenna [79]. (**b**) Wearable dual-band antenna fabricated on felt [80]. (**c**) Antenna fabricated on flexible silicone rubber substrate [81]. (**d**) Elliptically slotted patch antenna [82].

**Figure 9 micromachines-16-01028-f009:**
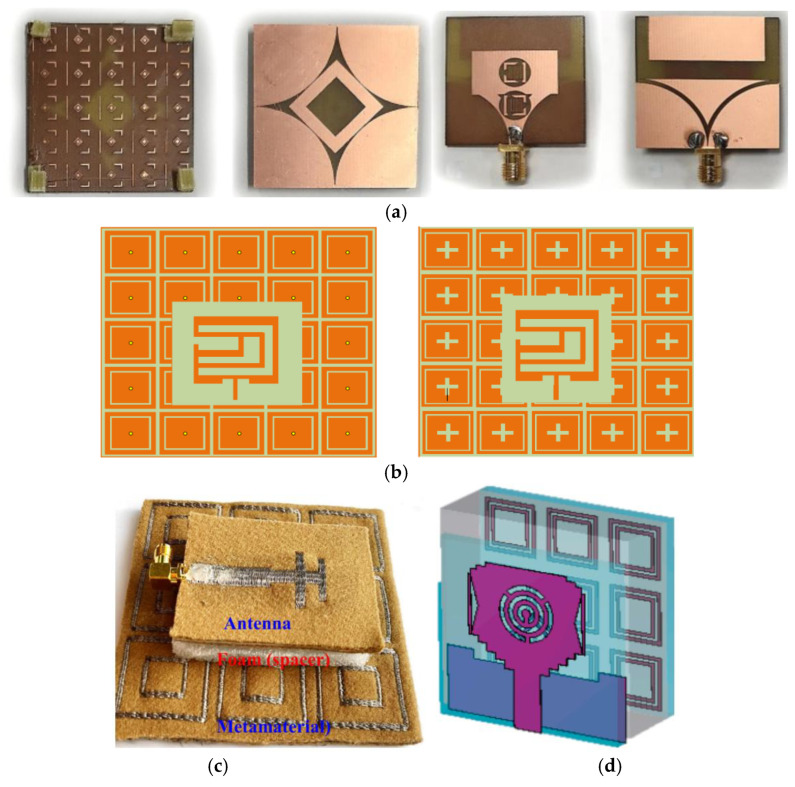
(**a**) Antenna prototype and AMC prototype [89]. (**b**) Mushroom-like EBG and cross-shaped EBG [90]. (**c**) Dual-band antenna with AMC structure fabricated on felt [91]. (**d**) Ultra-wideband antenna with AMC structure [92].

**Figure 10 micromachines-16-01028-f010:**
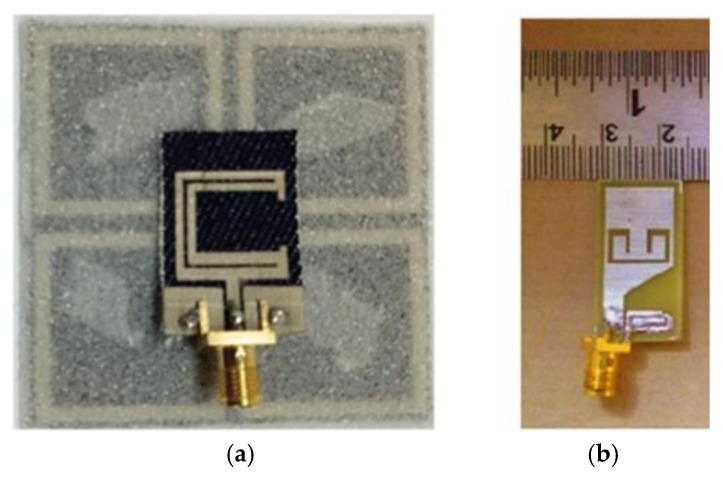
(**a**) CPW antenna integrated on an EBG structure [94]. (**b**) ACS antenna [95].

**Figure 11 micromachines-16-01028-f011:**
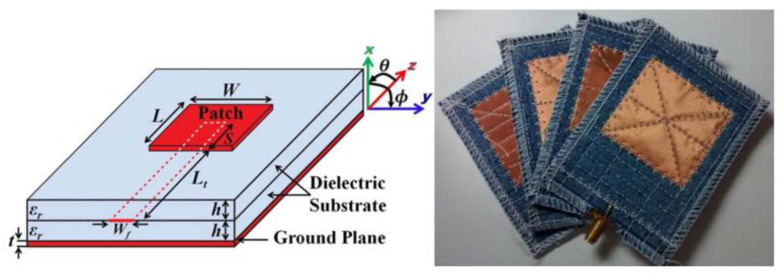
Textile patch antenna [96].

**Figure 12 micromachines-16-01028-f012:**
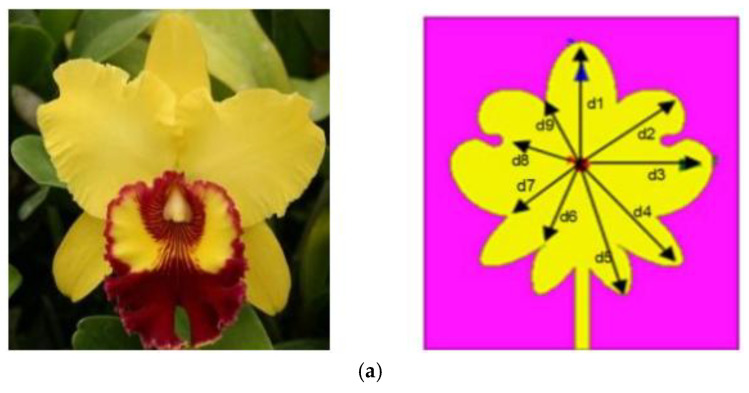
(**a**) Orchid flower and proposed antenna structure [101]. (**b**) Dual-band tree-shaped antenna [103]. (**c**) Biomimetic maple-leaf pod-shaped MIMO antenna [104].

**Figure 13 micromachines-16-01028-f013:**
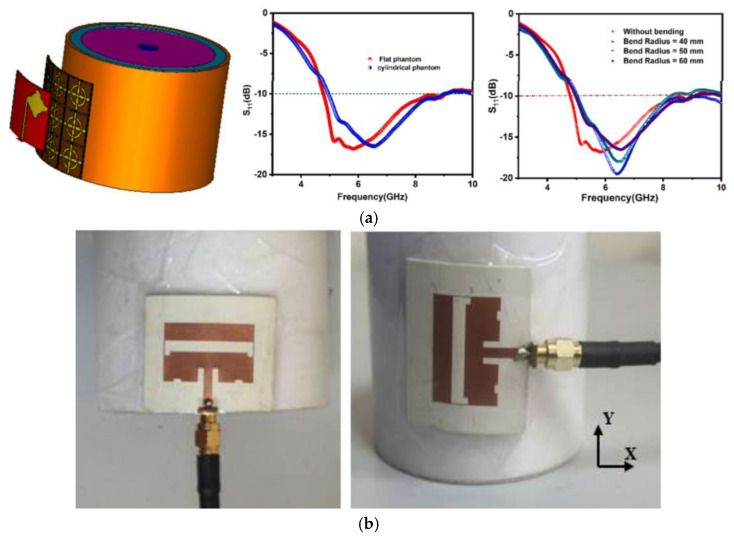
(**a**) Placing antenna integrated with FSS on the cylindrical phantom [109]. (**b**) Antenna bending along the X and Y axes [110].

**Figure 14 micromachines-16-01028-f014:**
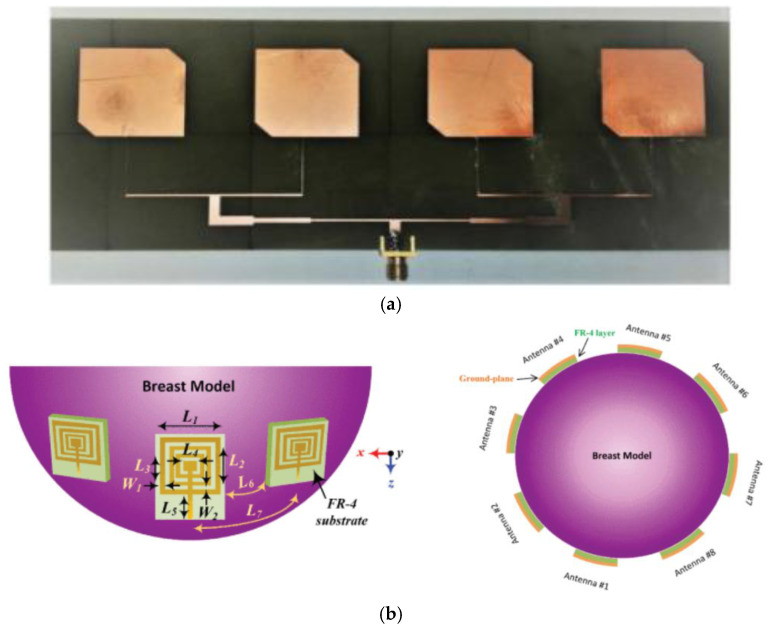
(**a**) Reconfigurable antenna array [121]. (**b**) Antenna array and location of the breast model on which the antenna array is placed [122].

**Figure 15 micromachines-16-01028-f015:**
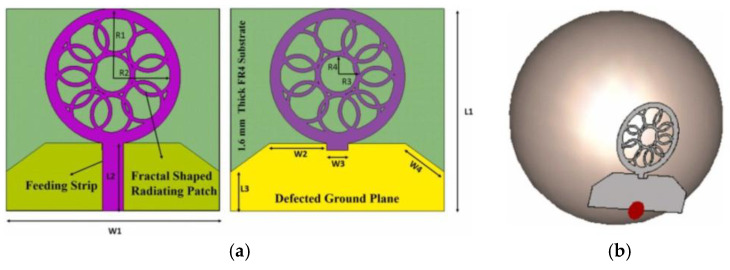
(**a**) UWB fractal antenna; (**b**) proposed antenna with breast Phantom [125].

**Figure 16 micromachines-16-01028-f016:**
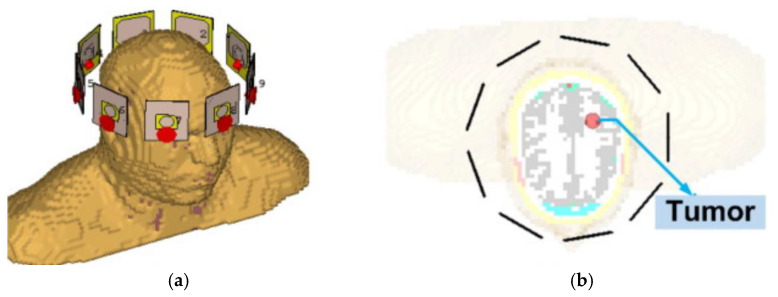
(**a**) Array-based simulation setup with 9-antenna element; (**b**) simulation setup with tumor-bearing Hugo head [126].

**Figure 17 micromachines-16-01028-f017:**
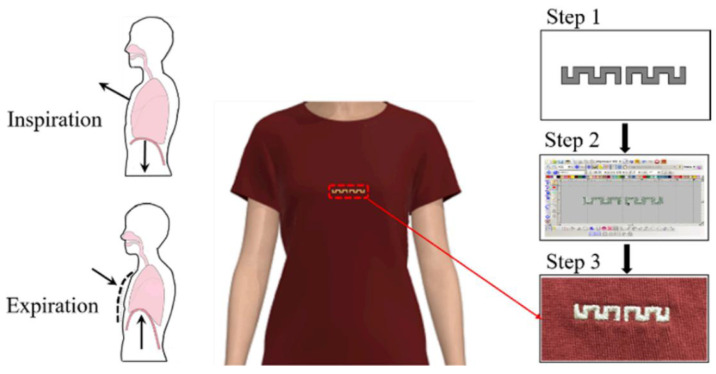
Simplified view of lungs during respiration and the embroidered dipole antenna [127].

**Figure 18 micromachines-16-01028-f018:**
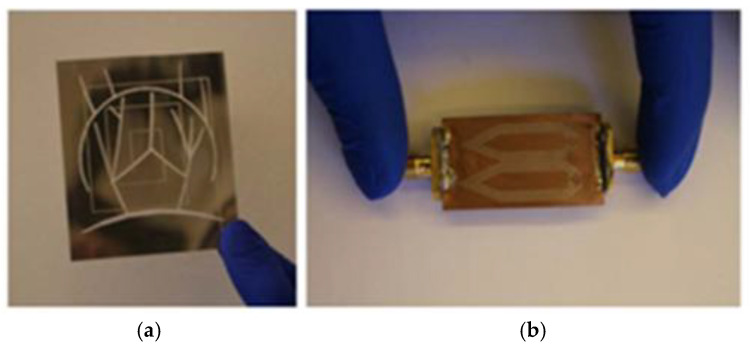
(**a**) Top sensing layer of the flexible slot antenna prototype; (**b**) top sensing layer of the band-reject filter prototype [131].

**Figure 19 micromachines-16-01028-f019:**
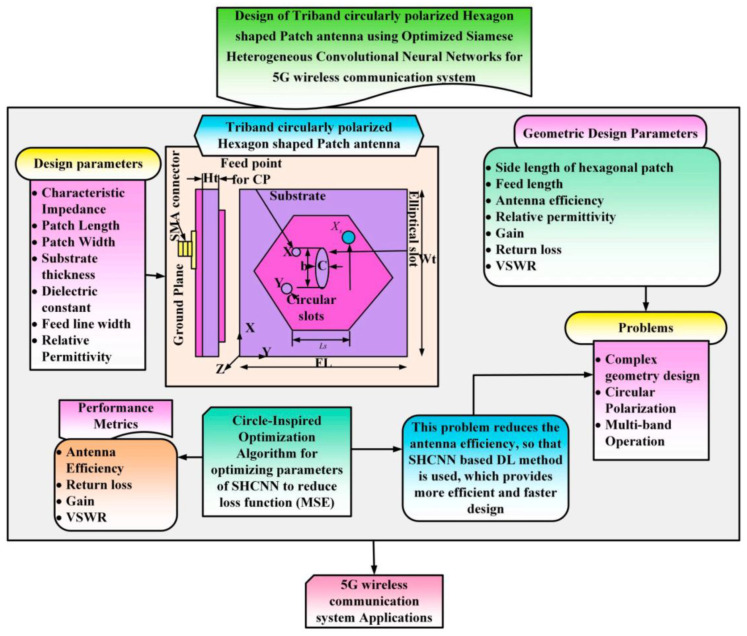
Design flow of the tri-band circularly polarized hexagonal patch antenna [147].

**Table 1 micromachines-16-01028-t001:** Conductivity and thickness of relevant conductive materials.

Conductive Material	Conductivity σ (S/m)	Thickness t (mm)
Copper [17]	5.9 × 10^7^	0.035
Aluminum [18]	3.8 × 10^7^	0.01
Stainless steel [19]	1.0 × 10^6^	0.45
Silver ink [24]	1.97 × 10^6^	0.12
Graphene [26]	1.1 × 10^6^	0.028
Copper–polyester taffeta [27]	0.4 × 10^5^	0.08

**Table 2 micromachines-16-01028-t002:** Parameters of relevant substrate materials.

Dielectric Material	Dielectric Constant (ε)	Dielectric Loss (tanδ)	Frequency Range
PET [28]	2.5	0.01	2.025–2.29 (GHz)
FR4 [29]	4.4	0.025	10.14–10.94 (GHz)
Denim [34]	1.7	0.024	2.4–14.88 (GHz)
Felt [35]	1.3	0.044	2.4–2.5 (GHz)
Rubber [37]	3.1	0.02	2.4–2.5 (GHz)
PTFE [44]	2.2	0.002	2.4 (GHz)
Synthetic paper [48]	2	0.0022	1.83–2.9 (GHz), 3.4–3.6 (GHz), 4.6–5.86 (GHz)
PI [49]	3.5	0.008	3.62 (THz)
PET paper [51]	3.2	0.022	4.01–5.05 (GHz), 6.23–7.58 (GHz)

**Table 3 micromachines-16-01028-t003:** Comparison of various bending angles used for wearable antennas.

Ref.	Substrate	Radius (mm)	Dimensions	Application
[90]	Polyethylene foam	(40, 60, 70)	68 × 73 × 3 (mm^3^)	WLAN
[91]	Felt	Rx (15, 30, 45) Ry (15, 30, 45)	0.41λ_0_ × 0.45λ_0_ × 0.016λ_0_	ISM
[94]	Jeans	Rx (70, 80, 100) Ry (70, 80, 100)	60 × 60 × 2.4 (mm^3^)	MBAN
[109]	Jeans	(40, 50, 60)	27 × 27.5 × 1 (mm^3^)	WBAN
[110]	Rogers 3003	Rx (50, 80, 100) Ry (50, 80, 100)	41 × 44 × 1.52 (mm^3^)	WBAN
[111]	Denim	(25, 35, 45)	60 × 60 × 1 (mm^3^)	ISM
[112]	Rogers 3003	Rx (40, 60, 80) Ry (25, 50, 75)	11 × 11 × 0.25 (mm^3^)	MM-WAVE

**Table 4 micromachines-16-01028-t004:** Performance parameters of wearable antennas for practical applications.

Ref.	Antenna Type	Frequency	Dimensions	Dielectric Materials	Peak of Gain	Bandwidth	Application
[114]	UWB antenna	3.1–10.6 (GHz)	60 × 60 (mm)	Felt	4.5 (dBi)	8.5 GHz	Breast cancer detection
[115]	UWB antenna	8.2–13 (GHz)	36 × 48 (mm)	Felt	7.04 (dB)	4.8 (GHz)	Skin cancer detection
[116]	Antenna array	2.4 (GHz)	200 × 78 (mm)	FR4	6.6 (dBi)	1 (GHz)	Kidney cancer detection
[117]	H-type patch antenna	5.4 (GHz)	50 × 50 (mm)	RT/duroid® 5880	5.25 (dB)	-	Thyroid cancer detection
[118]	U-shaped textile antenna	5.4 (GHz)	0.41λ_0_ × 0.54λ_0_	Non-woven polyester fabric	9.17 (dB)	952 (MHz)	Thyroid cancer detection
[119]	UWB stacked micro-strip antenna	6.1–12.3 (GHz)	36 × 30 (mm)	FR4	6.3 (dB)	6.2 (GHz)	Skin cancer detection
[121]	Reconfigurable antenna array	2.4 (GHz)	200 × 78 (mm)	FR4	3.1 (dB)	90.9 (MHz)	Brain cancer detection
[123]	Flexible antenna sensor	4.1 (GHz)	12 × 20 (mm)	PI	1.72 (dBi)	-	Blood glucose detection
[125]	UWB fractal antenna	3–10.2 (GHz)	28.5 × 20 (mm)	FR4	3.2 (dBi)	6.8 (GHz)	Breast cancer detection
[127]	Dipole antenna	2.4 (GHz)	45 × 4.8 (mm)	Cotton	1.86 (dB)	-	Breath detection
[133]	UBW antenna	3.15–10.55 (GHz)	16 × 10 (mm)	Polyimide	4.2 (dBi)	7.4 (GHz)	Health monitoring
[134]	AMC antenna	2.37–3.12 (GHz)	81 × 81 (mm)	FR4	6.2 (dBi)	750 (MHz)	Pulmonary edema monitoring
[135]	Textile antenna senor	2.4 (GHz)	40 × 50 (mm)	Thermoplastic polyurethane	-	-	Breathing monitoring
[138]	AMC antenna	23–27 (GHz)	9.6 × 14.1 (mm)	RO3003	6.73 (dBi)	4 (GHz)	Electronic travel aid

**Table 5 micromachines-16-01028-t005:** Comparison of related algorithms in the literature.

Ref.	Method	Advantages	Conclusion	Application
[146]	5-layer DNN model + Mixed Gravitational Search-Particle Swarm Algorithm	MGSA-PSO optimizes feeding phases	Combination of DNN and hybrid algorithms enables efficient beamforming	5G mm-Wave communication systems
[147]	Siamese Heterogeneous Convolutional Network + Circle-Inspired Optimization Algorithm	Shorten the design time	Algorithm combined with heterogeneous network for efficient multi-band circular polarization design	Multi-band communication for 5G base stations/UE devices
[148]	Ensemble Boosted Tree + Trilayered Neural Network + Fine Tree	Simultaneously optimize AR and S11 using machine learning methods.	Machine learning predictions align with simulation and actual measurement data.	Satellite and radar applications
[149]	GST-HBA algorithm: Tent chaos initialization + Golden Sine mechanism	Multi-function validation for optimization accuracy	Algorithm improvement significantly boosts optimization efficiency for complex electromagnetic problems	Antenna design parameter optimization
[150]	Adaptive NSGA-III multi-objective optimization algorithm	Pseudo-weight NSGA-III enhances Pareto front diversity	Data-driven modeling combined with optimization shortens design cycles, suitable for rapid iteration of complex parameters in high-frequency bands	Fast multi-objective optimization for 5G/6G mm-Wave antennas

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
