# Peer review of "Advancements in Wearable Antenna Design: A Comprehensive Review of Materials, Fabrication Techniques, and Future Trends in Wireless Communication"

_micromachines, 2025, doi:10.3390/mi16091028_

Round 1

Reviewer 1 Report

Comments and Suggestions for Authors

The review article is well structured. The authors tried to outline the main trends observed in the development of antennas for wearable electronics in recent years. The article includes tabular data summarizing the key specifications of flexible antennas and materials that will be interesting and informative for readers. Nevertheless, upon reviewing the document, I formulated several observations concerning the article.

1) Integrate references within the figure legends.

2) Organize the appended list (Line 910) of abbreviations into alphabetical order.

3) In accordance with sentence Line 195-197 the wrong figure presented in Fig. 2

4) Lines 194. The abbreviation for PET has not been introduced.

5) Lines 113-114. “ …Federal Communications Commission (FCC) officially allocated the spectrum from 3.1 GHz to 10.6 GHz for license-free UWB communication applications” in US only?

6) Line 177. Typo – “…Cost”.

7) In part 3.1.2 (in particular in Table 2) the parameter values of different substrate materials are presented. Indicate the frequencies utilized for measuring these parameters or the work frequency ranges within which these materials were applied in corresponding articles.

8) Typos in Line 294 (“toother”), Line 303 and in Line 332

9) Lines 461 -462. The sentence is not shortened in the text.

10) In Line 474 AMC abbreviation used wile it was introduced in Line 484 only.

11) Lines 502, 511, 521, 537 An inaccuracy in the enumeration of sections has been identified.

12) Please rectify the notation within formula 2 and include a citation to the relevant literature. If the formula is original, then provide an explanation of its derivation's underlying basis.

13) In some places, your review lacks the antenn parameters that you describe. You provide a description of the antenna and a reference, but do not provide parameters characterizing it (gain, frequency range, dimensions, substrate parameters, etc.).

14) It is not entirely clear why metamaterials are discussed in the chapter on antenna size reduction. Furthermore, several structures detailed within this section of the research do not exhibit the characteristics of metamaterials; consequently, they ought to be categorized as frequency-selective surfaces.

15) In Part 4 of the article, only sections 4.4 and 4.5 are of interest and should be expanded. Subsections 4.1 through 4.3 ought to be expunged from this section, as the characterization of antennas prior to section 4 already incorporates the parameters of gain, return losses and operational frequency range.

16) Sometimes references are given to works devoted to optical systems, for example [113]. No need to mix optical and microwave devices. It is better to remove them from review or form a separate section in the paper.

17) There are a lot of typos in the article, some of which I marked in my comments, but the article needs to be carefully re-read and edited by the authors.

Reviewer 2 Report

Comments and Suggestions for Authors

This paper presents a review of the design, fabrication and applications of wearable antennas, focusing on material selection, manufacturing and miniaturization techniques, feed types, performance and various application fields. An interesting section on the origins and history of wearable antennas is included. Design and analysis techniques and future trends are discussed.

It should be noted that several recent reviews on related topics are already available in the literature, e.g.

  • Yang, M.; Ye, Z.; Ren, Y.; Farhat, M.; Chen, P.-Y. Materials, Designs, and Implementations of Wearable Antennas and Circuits for Biomedical Applications: A Review. Micromachines 2024, 15, 26. https://doi.org/10.3390/mi15010026
  • Singh, S.; Mishra, R.; Kapoor, A.; Singh, S. A Comprehensive Review and Analysis of the Design Aspects, Structure, and Applications of Flexible Wearable Antennas. Telecom 2025, 6, 3. https://doi.org/10.3390/telecom6010003

The authors should cite these reviews and perform a content comparison to avoid unnecessary duplication. Complementarity in terms of material and aspects covered would be in favor of acceptance of the paper; it should be addressed by the authors.

In my opinion, the following sections / subsections should be improved or restructured:
"3.1.1. Conductive Materials": In this section, two specific examples of flexible antennas are discussed in detail; why they were selected and to what extent they are representative of the various cases of conductive materials employed should be clarified. Also, a discussion of the main features, merits and disadvantages, of the three groups of materials mentioned in the Section (as done e.g. in Section 3.1.2 for substrates), would be useful.
"4. Radiation Performance of Wearable Antennas": The section, as is, seems somewhat unbalanced, or even pointless. Subsections 4.1-4.4 contain little more than standard definitions of well-known concepts, while 4.5 references a couple of studies on the influence of bending but lacks a more general discussion. I think that the whole section, as it is, could be removed at almost no loss to the paper. Alternatively, it could be enriched, e.g. discussing parameters of gain, bandwidth etc. achieved for various state-of-the-art wearable antenna technologies (and perhaps the impact of various design factors). On the other hand, gain and bandwidth data for various antennas are already tabulated in Table 3 of Sec. 5.3, so such a discussion would also be appropriate there. If Sec. 4 is to be retained, a significant enrichment beyond what is already contained in Sec. 5.3 is required; if not, any standard definitions deemed necessary could be incorporated at other points as required.

Comments on the Quality of English Language

I feel that, in various points, the language and expression needs some improvement. A few (more or less random, but indicative) examples follow:
lines 144-145: "...antenna designs began to be compatible with other wireless technologies, such as 5G..." --> perhaps something like "...antenna designs were increasingly adapted to wireless technologies..." would more precisely convey the meaning
line 152: "In the year 2020 ..." --> perhaps "Since 2020 ..." or "From 2020 onwards ..." is better
lines 155-156: "... it is imperative to consider the shielding and influence of the human body ..." --> I think "shielding" is redundant here, "... the influence of the human body ..." suffices to cover everything.
lines 157-159: "such as frequency response, gain, and radiation efficiency, have prompted by response, gain, and radiation efficiency, have prompted ongoing optimization in their design." --> duplicated phrase, should be "such as frequency response, gain, and radiation efficiency, have prompted ongoing optimization in their design."
lines 163-164: "wearable antennas must be given greater consideration in the design of wearable antennas" --> meaningless sentence, needs clarification, perhaps "wearable materials..."?
lines 461-462: "... the concept of digital metamaterials, in which the electromagnetic properties." --> unfinished (abruptly terminated) sentence
line 749: "5.3. Another Applications" --> should read "Other Applications" (or something like this)

Round 2

Reviewer 1 Report

Comments and Suggestions for Authors

All corrections in the article were made in accordance with the comments provided in the previous review report. However, the final version of the article needs to be edited because some of the figures (for example, Figure 7) are not displayed and many references to literature are not displayed in the text ("Error! Reference source not found"). I would advise the authors to check the article before submitting it for re-review. And when you respond to comments on the review, indicate the line numbers from the final version of the article. In the document with the responses to the review that you provided, the line numbers are not correct.

Author Response

Comments 1: All corrections in the article were made in accordance with the comments provided in the previous review report. However, the final version of the article needs to be edited because some of the figures (for example, Figure 7) are not displayed and many references to literature are not displayed in the text ("Error! Reference source not found"). I would advise the authors to check the article before submitting it for re-review. And when you respond to comments on the review, indicate the line numbers from the final version of the article. In the document with the responses to the review that you provided, the line numbers are not correct.

Response 1: Thank you for pointing out this issue. My local Word file does not display the error message “Error! Reference source not found,” and the image captions appear correctly. We have re-uploaded the article in both Word and PDF formats and corrected the erroneous line numbers.

Reviewer 2 Report

Comments and Suggestions for Authors

After the revisions carried out by the authors, I would like to suggest publication. But please notice that something seems to have gone wrong with the formatting in the revised paper, resulting in many instances of the "Error! Reference source not found" message in the text.

Author Response

Comments 1: After the revisions carried out by the authors, I would like to suggest publication. But please notice that something seems to have gone wrong with the formatting in the revised paper, resulting in many instances of the "Error! Reference source not found" message in the text.

Response 1: Thank you for pointing out this issue. My local Word files do not display the error message “Error! Reference source not found”, We have re-uploaded the article in Word and PDF formats.
